



# Observations of Particles at their Formation Sizes in Beijing, China

Rohan Jayaratne[1†], Buddhi Pushpawela[1†], Congrong He[1], Jian Gao[2], Li Hui[2], Lidia Morawska[1*],

7   1   International Laboratory for Air Quality and Health, Queensland University of
8       Technology, GPO Box 2434, Brisbane 4001, Australia.
9   2   Chinese Research Academy of Environmental Sciences, Beijing 100012, China.

† Joint first authors.
* Corresponding author contact details:
Tel: (617) 3138 2616; Fax: (617) 3138 9079
Email: l.morawska@qut.edu.au





25                                            **Abstract**

New particle formation (NPF) has been observed in many highly polluted environments of South-
East Asia, including Beijing, where the extent of its contribution to intense haze events is still an
open question. Estimated characteristics of NPF events, such as their starting times and formation
and growth rates of particles, are very different when the measurements are restricted to particles
in larger size ranges. In order to understand the very first steps of particle formation, we used a
neutral cluster and air ion spectrometer (NAIS) to investigate particle characteristics at sizes exactly
where atmospheric nucleation and cluster activity occurs. Observations over a continuous three-
month period in Beijing showed 26 NPF events. These events generally coincided with periods with
relatively clean air when the wind direction was from the less-industrialized north. No NPF were
observed when the daily mean $PM_{2.5}$ concentration exceeded 43 $\mu g\ m^{-3}$, which was the upper
threshold for particle formation in Beijing.  The fraction of particles that are charged in the size range
2-42 nm was normally about 15%. However, this fraction increased to 20-30% during haze events
and decreased to below 10% during NPF events. With the NAIS, we determined the starting times of
NPF very precisely to a greater accuracy than has been possible in Beijing before and provided a
temporal distribution of NPF events with a maximum at about 8.30 am. Particle formation rates
varied between 10-36 $cm^{-3}\ s^{-1}$.  Particle growth rates were estimated to be in the range  0.5-9.0 nm
$h^{-1}$. These results are more reliable than previous studies in Beijing as the measurements were
conducted for the first time at the exact sizes where clusters form into particles and provide useful
insight into the formation of haze events.


**Keywords:**  New particle formation, secondary particles, nucleation, haze events
______________________________________________________________________



## 1. Introduction

Particles in the atmosphere may be classified into two types depending on their origin. Primary particles are directly emitted by a source while secondary particles are formed through a secondary process by the homogeneous condensation of gaseous precursors. This is known as new particle formation (NPF) and has been observed in many parts of the world in many different types of environment (Curtius, 2006;Kulmala et al., 2005;Kulmala et al., 2004;Zhang et al., 2011). NPF is a complicated process where molecular clusters come together to form particles at a size of about 1.6 nm (Kulmala et al., 2004). Generally, it is favoured by clean air conditions where the particle number concentration (PNC) in the atmosphere is low, resulting in a lower particle surface available for the condensation of gases, leading to an increase of the supersaturation in the air enhancing homogeneous condensation of the gaseous species (Kulmala et al., 2005;Wu et al., 2011) and, therefore, NPF is less frequent in polluted environments. However, if the gaseous precursor concentration is high enough, NPF may occur at even higher particle concentrations (Kulmala et al., 2005;Wu et al., 2011). Jayaratne et al. (2015) showed that in the relatively clean environment of Brisbane, Australia, NPF do not occur when the ambient $PM_{10}$ concentration exceeds about 20 µg $m^{-3}$. However, NPF have been commonly observed in more polluted environments like Beijing (Kulmama et al., 2016) and Shanghai (Xiao et al., 2015) in China. The study of the formation and characteristics of NPF events in Beijing is important because of its possible influence on severe haze episodes (Guo et al., 2014;Huang et al., 2014). Such haze events not only give rise to poor visibility but are responsible for sharp increases in respiratory problems amongst the large population that is exposed. In particular, Beijing experienced severe haze episodes during November and December, 2015. Daily maximum $PM_{2.5}$ values in the city exceeded 500 µg $m^{-3}$ on no less than six days during the month of December, prompting two official air pollution 'red alerts' to be issued (Xue et al., 2016). Close examination of the haze events demonstrate that they occur in cycles of a few days and generally coincide with winds blowing from the more polluted regions south of the city (Guo et al.,



2014;Wu et al., 2007). Particulate matter concentrations are observed to drop significantly when the
winds change to a northerly direction, bringing cleaner air into the city, which is when NPF events
generally occur (Guo et al., 2014).

The earliest study of NPF using a TSI scanning mobility particle sizer (SMPS) in Beijing was carried out
by Wehner et al. (2004). They observed NPF on 25 out of 45 days of measurement with PNCs
exceeding $10^5$ cm$^{-3}$. Subsequent studies using the SMPS were carried out by Wu et al. (2007) who
showed that NPFs were observed on 50%, 20%, 35% and 45% of days during the spring, summer, fall
and winter seasons, respectively. Yue et al. (2010) investigated 12 NPF events and showed that
sulphuric acid and ammonia accounted for about 45% of the growth rate, with the balance being
due to organic species. Guo et al. (2014) conducted a detailed analysis over a two-month period
during the fall of 2013 and showed that NPF events occurred in a clear periodic cycle of about 4-7
days coinciding with northerly winds bringing cleaner air into the city. The average PM$_{2.5}$ values
when the wind was from the north and when it was from the south were 35 and 114 µg m$^{-3}$,
respectively. The average PM$_{2.5}$ (and PNC) values during and outside the NPF periods were less than
50 µg m$^{-3}$ (greater than 2 x $10^5$ cm$^{-3}$) and several hundred µg m$^{-3}$ (5 x $10^4$ cm$^{-3}$), respectively. Pollution
also originates from within the city – from motor vehicle emissions and industrial sources. In general,
airborne gaseous pollutants in Beijing and other urban regions in China are mainly volatile organic
compounds (VOC) and oxides of nitrogen (NO$_x$) from local transportation and sulphur dioxide (SO$_2$)
from regional industrial sources (Wang et al., 2009;Yue et al., 2010). However, Guo et al. (2014)
showed that the nucleation and growth processes occurred on a regional scale, over several
hundred km, with the effect of local sources such as motor vehicle emissions being insignificant. A
good summary of the studies conducted since 2004 in Beijing may be found in Zhibin et al. (2013)
and Kulmama et al. (2016).



All these previous studies in Beijing have been carried out using the SMPS. The SMPS is a good tool
to determine the PNC and size distribution down to a minimum particle size of about 3 nm, although
the efficiency of detection falls off below about 10 nm. Thus, an event where aerosols in the size
range 3-10 nm emitted on-site as primary particles or entrained from a distant location that
continue to grow to larger sizes may be mistaken for particle formation at that monitoring site. The
SMPS is also not able to identify the exact time period during which particle formation occurs. An
instrument that can detect particles at smaller sizes is the neutral cluster and air ion spectrometer
(NAIS) from Airel Ltd. The NAIS is specifically designed to monitor particle formation as it can detect
particles down to a size of 0.8 nm (Manninen et al., 2016;Manninen et al., 2009;Mirme et al., 2007).
In this paper, we present the first results of using a NAIS in Beijing over the course of three months,
two months with intense haze and very few NPF events, and the other including several days with
NPF. We will investigate the characteristics of the NPF events and the conditions that gave rise to
them. As the measurements included the sizes at which particles formed, the results provide more
reliable information of such parameters as the starting times, growth rates and formation rates of
particles than has been possible in the past.

**2.  Methods**

**2.1 Instrumentation**
The NAIS is an improved version of the air ion spectrometer (AIS) which was developed by Airel Ltd
(Mirme et al., 2007). In both instruments, the sample air is split equally into each of two separate
cylindrical spectrometer columns, one of each polarity. At the inlet to each column, a unipolar
corona wire diffusion charger of the same polarity as the central electrode in the column brings the
particles to an equilibrium charge distribution. They are then classified by a differential mobility
analyser where the outer electrodes consist of 21 insulated sections or rings, each with its own
electrometer.   The charged particles in the air flow are repelled by the central electrode which has a



tapered cross-section and collected by the rings. The electric field between the central electrode and
the rings is fixed by the voltage on the inner electrode and the gap between the inner and outer
electrodes so that only particles in a given mobility range may be collected by each ring. In this way,
the instrument can separate particles into 21 mobility or size bins. A refinement in the NAIS over the
AIS is that it uses controlled charging to measure the concentration of charged particles in addition
to the total PNC in each size range.  This is done by switching the voltage off on the corona charger
during one part of the measurement cycle. Thus, the NAIS can measure both charged and neutral
particles separately. The mobility range of the instrument is 3.16-0.001 $cm^2 V^{-1} s^{-1}$ which corresponds
to a mobility diameter range of 0.8-42 nm.  A good description of the detailed operation of the NAIS
may be found in Manninen et al. (2016). In this study, we set the NAIS to a measurement cycle of 5
min consisting of 2 min each for charged and neutral particles with an offset period of 1 min. Thus, a
PNC and charged particle concentration reading were obtained in real time once every 5 min.

The larger size PNC was monitored with an SMPS. The instrument was set to scan up and retrace
times of 120 and 15 s respectively. The aerosol and sheath flow rates were 0.3 and 3.0 lpm,
respectively. Size distributions were determined in 107 bins in the size range 14 to 673 nm. A
complete size distribution record was obtained every 5 min. $PM_{2.5}$ concentrations were monitored
with a tapered element oscillating monitor (TEOM) and recorded as hourly average values.

**2.2 Study Design**
The NAIS and SMPS were set up within a room on the roof of the Chinese Research in Atmospheric
and Environmental Sciences (CRAES) Building in Beiyuan, Beijing, on the 28 October 2015 and
monitoring was conducted continuously until 31 January 2016. This comprised 96 days including
several episodes of very high pollution or haze days when the $PM_{2.5}$ in Beijing exceeded 100-200 µg
$m^{-3}$. Owing to the high PM content in the air, the instrument experienced some problems on 9 days
during which data was lost. Air was sampled through a straight steel pipe of diameter 4 cm
protruding vertically through the roof of the building. Meteorological parameters, including the wind
speed, wind direction, air temperature and relative humidity were monitored and recorded hourly
over the course of the study period.

**2.2 Analysis**

**2.2.1 Identification of NPF events**
The NAIS provided spectrograms showing the neutral and charged particle number size distributions
in real time with the concentrations shown in colour contours. The neutral and charged PNCs were
also provided in real time at 5 min intervals. NPF events were identified using the method proposed
by Zhang et al. (2004). We calculated the rate of change of PNC, dN/dt, where N is the number of
particles in the size range 1.8 -10.0 nm. Events with N > 10,000 cm$^{-3}$ for at least 1 hour and dN/dt
>15,000 cm$^{-3}$h$^{-1}$ were classified as NPF events. These events generally exhibited a 'banana shape' in
the spectrograms. A day on which there was at least one NPF event as defined above was termed an
"NPF day". A day where the above criteria were not fulfilled were classified as a "non-event" day.
NPF events are characterised by sharp increases in the intermediate size range.  The starting times of
an event was determined by using the time of sudden increase in PNC in the size range 1.8 – 10.0
nm.

**2.2.2 Condensation sink (CS) and coagulation sink (CoagS)**
The condensation sink of particles is defined as (Dal Maso et al., 2002;Dal Maso et al., 2005;Kulmala
et al., 2012;Lehtinen et al., 2003;Salma et al., 2011)
$$CS = 2\,\pi\,D\,\sum_i \beta_m\,(d_{p,i})d_{p,i}N_i$$

(1)





where $D$ is the diffusion coefficient of the condensing vapour and  $\beta_m$ is the transition correction
factor for mass flux. $d_{pi}$ and $N_i$ are the diameter and the number concentration of particles in the size
bin $i$ respectively. The unit of CS is s$^{-1}$.
Assuming that the main condensing vapour is sulphuric acid, we estimated the diffusion coefficient
for condensing vapour using the expression
$D = 5.0032 * 10^{-6} + 1.04 * 10^{-8}T + 1.64 * 10^{-11}T^2 - 1.566 * 10^{-14}T^3$          (2)
where $D$ has the units of m$^2$ s$^{-1}$ and where the temperature $T$ is in Kelvin (Jeong, 2009).
The transition correction factor, $\beta_m$, was calculated using the Fuchs-Sutugin expression (Fuchs and
Sutugin, 1971)

$$\beta_m = \frac{Kn + 1}{1 + (\frac{4}{3\alpha} + 0.337)Kn + (\frac{4}{3\alpha})Kn^2}$$

(3)

where

$Kn = \frac{2\lambda}{d_p}$      and      $0 \le \alpha \le 1$ .


Here, $Kn$, the Knudsen number, describes the nature of the suspending vapour relative to the
particle, $\lambda$ is the mean free path of a suspending vapour molecule and $d_p$ is the diameter of the
particle (Seinfeld and Pandis, 2006). The mass accommodation coefficient (sticking coefficient) $\alpha$
describes the probability of a vapour molecule sticking to the surface of a particle during vapour-
particle interactions (Seinfeld and Pandis, 2006). In this study, we assumed $\alpha = 1$.

The relationship between the condensation sink and coagulation sink is given by Lehtinen et al.
(2007) as





$$CoagS_{d_p} = CS.\left(\frac{d_p}{0.71}\right)^m$$

(4)

where the exponent m varies in the range -1.75 to -1.5 with a mean value -1.7 and the value 0.71 is
the diameter of a hydrated sulphuric acid molecule. The unit of CoagS is $s^{-1}$.

In order to calculate the CS, we used the PNC obtained from the SMPS in the 107 size bins. We
calculated $D$ using equation (2) at temperature T = 303 K. The values used for the exponent m was
-1.7 (Dal Maso et al., 2008) and λ =108 nm (Massman, 1998).

**2.2.3 Particle formation rate**
Particle formation or nucleation occurs from thermodynamically stable clusters in the size range 1.0-
2.0 nm (Kulmala et al., 2007). The formation rate may be estimated from the number of particles in
the smallest size bin, usually 2-3 nm in the NAIS.
The formation rate of particles is defined as

$$J_{d_p} = \frac{dN_{d_p}}{dt} + CoagS_{d_p}.N_{d_p} + \left(\frac{GR}{\Delta d_p}\right)N_{d_p}$$

(5)

where $N_{dp}$ is the number concentration of particles in the size range $d_p$ and ($d_p$ + $\Delta d_p$) respectively
(Kulmala et al., 2012). In this study, we used the values $d_p$ = 2 nm and $\Delta d_p$ = 1 nm, corresponding to
the size range 2-3 nm. $CoagS_{dp}$ represents the loss of the particles due to coagulation and $GR$ is the
growth rate of particles. The unit of formation rate is $cm^{-3}\ s^{-1}$.





### 2.2.4 Particle growth rate (GR)


During an NPF event, the growth rate of particles was defined by Kulmala et al. (2012) as

$$GR = \frac{dd_p}{dt} = \frac{d_{p2} - d_{p1}}{t_2 - t_1}$$

(6)

where $dp_2$ and $dp_1$ are the diameters of particles at times $t_2$ and $t_1$, respectively. This was calculated
by the maximum concentration method as described in Kulmala et al. (2012) by examining the time
of maximum PNC at each particle size during an NPF event. First, we exported the number
concentrations of particles obtained from the NAIS in 15 bins in the size range 1.8 – 42.0 nm. Next,
we selected the time of maximum concentrations during each NPF event for each particle size bin.
Finally, we calculated the growth rate using the slope of the best-fitted line on the graph of median
diameter of particle in each size bin vs. the time of maximum concentration. The unit of GR is nm h$^{-1}$.

### 3. Results and Discussion



### 3.1 Distribution of NPF events


During the entire period of measurement, the NAIS yielded 87 complete days of data, the
remaining 9 days being affected by instrument faults, generally due to power fluctuations.
November and December 2015 were particularly prone to high pollution events in Beijing. The
daily average PM$_{2.5}$ concentration exceeded the recommended maximum of 50 μg m$^{-3}$ in Beijing
on 47 days during this two-month period. The maximum daily average was 448 μg m$^{-3}$ and this
occurred on 1$^{st}$ December. Owing to the high condensation sink on polluted days, there were
relatively few NPF days during these two months. There was a relative improvement of air
quality after 4$^{th}$ January and this lasted until 31$^{st}$ January - the end of the monitoring period,
during which time, the daily average exceeded 100 μg m$^{-3}$ on only four days. Enhanced PM$_{2.5}$



concentrations (> 50 µg m$^{-3}$ ) were observed on 15 days in January. These days occurred in
groups and we could identify five such distinct periods during January. No NPF events were
observed during these 15 days; however, several NPF events were observed on the other days
during the intervening periods. A summary of the observational days, together with the number
of days on which data were available and NPF events were observed, are shown in Table 1.
Column 3 shows the numbers of days on which complete 24-hour data were obtained. We note
that, during the 56 such days between 27[th] October and 31[st] December, NPF events were
observed on just 10 days, whereas during the 31 days in January 2016, NPF events took place on
16 days. The near equal division between NPF days and no-NPF days in January provided an ideal
data set to compare the parameters and conditions on these two types of days. The difference
between November/December and January had a clear dependence on the PM$_{2.5}$
concentrations. Figure 1 gives a summary of the days on which NPF events were observed.

**3.2 Relationship between NPF events and PM$_{2.5}$ concentration**
In Fig 2, we take a closer look at the January data, together with the respective mean daily PM$_{2.5}$
concentrations. It is apparent that there were five distinct groups of NPF days in January. These
are labelled in 2(b). In the NAIS spectrogram, shown in 2(a), the 16 NPF events are clearly
observed with the characteristic 'banana' shapes compressed into near-vertical bands extending
up from the smallest sizes. The five groups from left to right consist of 5, 3, 2, 5 and 1 NPF
events, respectively (Figs 2(a and b)). These groups are separated by time periods when no NPFs
were observed. The PM$_{2.5}$ values are clearly lower on NPF days than on the other days with
mean daily values of 18 µg m$^{-3}$ and 120 µg m$^{-3}$, respectively. A Student's t-test showed that the
difference in mean daily PM$_{2.5}$ values between NPF days and the other days was statistically
significant at the confidence level of 95%. Figure 2(c) shows the corresponding mean daily PNC.
While the PNC within each group showed a greater fluctuation than the PM$_{2.5}$, the PNC on NPF
days was significantly higher than on non-NPF days. Therefore, although the PM is higher on





haze days than on NPF days, the t-tests again showed that the PNC was significantly lower on
haze days than on NPF days. This is explicable in terms of the particle size. Particles are
significantly larger on haze days than on clean days when NPF events are likely to occur.

In Fig 3, we plot the daily mean PNC against the daily mean $PM_{2.5}$ for the 31 days in January. The
days with NPF and the days with no NPF events clearly fall into two distinct groups according to
the daily mean $PM_{2.5}$ values.  No NPF events were observed on a day when the mean $PM_{2.5}$ value
exceeded 43 µg m$^{-3}$. There is some minor overlap in the PNC values on the two types of days but
this is primarily because they are daily averages. When we consider the average PNC values
during the NPF events alone, a t-test showed that they are significantly higher than at other days
and times. However, we do see that, on haze days, the daily average PNC does not exceed 8.5 x
$10^{4}$ cm$^{-3}$.

**3.3 Relationship between NPF events and wind direction**
Previous studies have shown that the wind direction played an important role in determining the
$PM_{2.5}$ concentration in Beijing (Guo et al., 2014). Again, we look at the month of January, as it
provided an almost equal number of NPF days and non-NPF days and was, therefore, ideal to
compare the wind direction on the two types of days. Figure 4 shows the wind direction roses for
both NPF days and non-NPF days during January. The frequencies are given as percentages of time
when the wind was from a given direction. There is a clear difference between the two sets of days
with a strong correlation between the NPF days and the wind direction. NPF events clearly occurred
on days when the wind direction was predominantly from the NW, while it was more equally
distributed with a greater likelihood of arriving from the S and E during the haze days when there
were no NPF events. The frequencies in the sector between NW (315⁰) and N (0⁰) on NPF days and
non-NPF days were 68 % and 11 %, respectively.  Air from the north of Beijing is usually cleaner than
that from the more industrialized south of the city (Guo et al., 2014). Clean periods are characterised





by decreased condensation sinks that promote NPF. Winds from the south bring a copious supply of
freshly available gaseous precursors that should give rise to particle formation. However, the
absence of NPF events during these times suggests that the wind is also carrying a large supply of
particles that reduce the gaseous supersaturations required for particle formation. Thus, the
observed haze events are more likely to be due to particles carried by the wind into the city or being
prevented from escaping due to temperature inversions in the atmosphere.

**3.4 Charged particles and clusters**
Next, we look at the behaviour of charged clusters and charged particles, with particular attention to
NPF events and haze events. In order to compare and contrast the characteristics of these particles,
we selected a period of four days, comprising two haze days that were immediately followed by two
NPF days. Figure 5 shows the time series of the concentration of total and charged particles (a) and
clusters (b) observed over this four-day period from November 30 to December 3. In Fig 5(a), the
upper curve represents the total PNC while the lower curve gives the charged PNC. The difference
between the two curves gives the neutral PNC.  This is similar for the cluster concentrations in Fig
5(b). The conditions during the two types of events could be compared during this period as intense
haze was observed on the first two days (Nov 30 and Dec 1) while, following  a change of wind
direction near midnight on the 1 December,  two strong NPF events took place on the next two days
(Dec 2 and 3). In general, the neutral cluster concentration exceeded the cluster ion concentration
by about two orders of magnitude, with this ratio being somewhat greater when there was no
particle formation. Large pools of neutral clusters were always observed to be present in previous
studies in the boreal forests of Hyytiala, Finland (Kulmala et al., 2007) and in the urban environment
of Brisbane, Australia (Jayaratne et al., 2016). Here, we can confirm the same observation in the
more polluted Beijing atmosphere. The total cluster concentration showed a significant decrease, by
almost an order of magnitude, as we passed from the first two days to the two NPF days. We
attribute this to two phenomena - the attachment of clusters to existing particles and the conversion
of clusters to new particles. We also see that less than 10% of the particles were charged, both
during NPF days and when there were no NPF events.

A summary of the neutral and charged PNC and cluster concentrations during the various stages
over the entire period of observation are presented in Table 2. Also shown are the percentage
numbers of all particles that were found to be charged. NPF events and NPF days are defined in
section 2.2.1. A haze day was defined as a day when the 24-hour average $PM_{2.5}$ concentration
exceeded 75 μg m$^{-3}$ - the national air quality standard in China. A day that met neither of these
criteria was defined as a 'normal day'. Thus, by our ad-hoc definition, a normal day had a daily
average $PM_{2.5}$ concentration in the range 43-75 μg m$^{-3}$, since no NPF events were observed on days
when the average $PM_{2.5}$ concentration was greater than 43 μg m$^{-3}$. The duration of the various
events affected the daily values while the conditions during the events affected their peak number
concentrations. This introduced an inherent uncertainty of up to 20% in the values shown in the
table.

We note that only a very small percentage of clusters, less than 1%, are charged under all conditions.
On a normal day, around 15% of the particles larger than 2 nm are charged. The fraction that is
charged decreases significantly during an NPF event. This is consistent with our observations in
Brisbane (Jayaratne et al., 2016) and may be attributed to the rapid increase in particle number and
the associated coagulation. On the other hand, during a haze event, the percentage of particles
charged increases to a value between 20% and 30%. These observations are consistent with the PNC
and particle sizes and the equilibrium distribution of charge on particles. NPF are characterised by
large numbers of small particles while the SMPS and TEOM show that haze events comprise much
larger particles. The amount of charge that a particle can hold and the fraction of particles that are



charged in equilibrium both increase with particle size, so it is not unexpected to find that a larger
percentage of particles are charged during the haze events.

**3.5 Particle formation times**
All except one of the 26 NPF events during the period of observation began between 7:30 am and
10:00 am. The mean time was 8:45 am. This result is in agreement with Wu et al. (2007) who, using
an SMPS, reported that NPF events during clean air periods in November, December and January
generally started between 7:00 am and 10:00 am. Figure 6 shows the temporal distribution of the
start times of the NPF events, classified into 30 minute bins. The most likely time for an NPF event to
begin was between 8:00 and 8:30 am. This time coincides with the morning rush hour traffic when
the production rate of gaseous precursors is generally at a maximum. Sunrise in Beijing in
December/January is at about 7.30 am.

Figure 7 shows the NAIS spectragram of the strong NPF event that occurred on 29[th] October 2015.
The spectragram shows a clear banana profile which levels off at about 20 nm. The PNC in this event
was relatively high, exceeding $1.6 \times 10^5$ cm$^{-3}$ near 11:00 am. The PM$_{2.5}$ concentration remained
between 12 and 16 µg m$^{-3}$ right through this event. The markers shown on this figure are the median
sizes of particles at each time. In the spectragram, the transition time from clusters to particles, at
around 2 nm, is very sharp and we can conclude that particle formation began at around 09:00 h.
However, previous studies in Beijing have not been able to measure particles smaller than 3 nm. In
Fig 7, if we truncate the lower particle size margin to 3 nm, the starting time of the NPF event
appears later than it actually is, approximately at 9:30 am. In other NPF spectragrams, we see this
difference being as much as 1.0 to 1.5 h depending on the initial growth rate. Thus, we conclude that
the starting times that we have derived (Fig 6) are more accurate than has been obtained in the
past. This will also affect the estimated growth rates of particles during NPF events as we shall show
in the next section.




### 3.6 Condensation and coagulation sinks


The condensation and coagulation sinks were calculated during NPF events assuming the growth to

be due to sulfuric acid and using the SMPS data and the equations given in the methods section. The

mean value of the condensation sink was $5 \times 10^{-3}$ $s^{-1}$. This value is somewhat smaller than that

reported by Wu et al. (2007) ($1.4 \times 10^{-2}$ $s^{-1}$) and Wu et al. (2011) ($1 \times 10^{-2}$ $s^{-1}$) but within the range of 0 –

$5 \times 10^{-2}$ $s^{-1}$ reported in all NPF events between 2004 to 2008 in Beijing by Zhibin et al. (2013). The

mean value of our coagulation sink for 2 nm particles during an NPF event was $9 \times 10^{-4}$ $s^{-1}$. Previous

studies in Beijing have not been able to determine this value at 2 nm. The values reported for 3, 5

and 10 nm for NPF events in Beijing by Wu et al. (2011) are $9.9 \times 10^{-4}$ $s^{-1}$, $4.3 \times 10^{-4}$ $s^{-1}$ and $1.4 \times 10^{-4}$ $s^{-1}$,

respectively. The value at 3 nm is close to our value at 2 nm.

### 3.7 Particle formation rate

Using the values of the condensation and coagulation sinks in equation 5, we calculated the

formation rate of particles in the smallest particle size bin 2-3 nm where the rate of increase of

particles ranged from about $5.0 \times 10^{3}$ to $1.5 \times 10^{4}$ $cm^{-3}$ $h^{-1}$. The resulting formation rates varied

between 10 and 36 $cm^{-3}$ $s^{-1}$, with a mean of 23 $cm^{-3}$ $s^{-1}$.  Previous estimates in Beijing did not have

the benefit of the information in the 2-3 nm size bin. Wu et al. (2007) calculated the formation rate

in the wider size bin of 3-10 nm and arrived at a value in the range 3.3-81.4 $cm^{-3}$ $s^{-1}$ with a mean of

22.3 $cm^{-3}$ $s^{-1}$. Yue et al. (2010) studied 12 NPF events in Beijing and derived a formation rate in the

range 2-13 $cm^{-3}$ $s^{-1}$ and showed that the formation rate was directly proportional to the sulfuric acid

concentration. They did not specify the size range used in this calculation but the smallest detectable

particle size of the instrument used was 3 nm.






**3.8 Particle growth rate**

In the NPF event shown in Fig 7, the particle growth rate soon after formation is about 9 nm h$^{-1}$. The
average growth rate during the entire event (between 9:00 and 11:00 am) estimated from equation
6 was 4.8 nm h$^{-1}$. Although the PNC reached very high values, the particles did not grow much larger
than about 30 nm, suggesting that the high condensation sink was restricting the precursor gas
concentration in the atmosphere. The growth rate of all the NPFs observed ranged from 0.5 to 9.0
nm h$^{-1}$ with a mean value of 3.5 nm h$^{-1}$. Previous estimates of the growth rate during NPF using the
SMPS have yielded mean values of 1.0 nm h$^{-1}$ (Wehner et al., 2004) and 1.8 nm h$^{-1}$ (Wu et al., 2007).,
2007). Zhibin et al. (2013) determined the growth rates of a number of NPFs in Beijing over a 4-year
period and reported a range of 0.1 to 10 nm h$^{-1}$ with a mean of 3.0 nm h$^{-1}$ which is in close
agreement with our value.





**4.   Summary and Conclusions**
We monitored charged and neutral PNC over a continuous three-month period for the first time in
Beijing. The results showed 26 NPF events. No NPF were observed when the daily mean $PM_{2.5}$
concentration exceeded 43 μg m$^{-3}$.
A summary of the main parameters determined are shown in Table 3.
This is the first study of NPF in the particle size range below 3 nm in Beijing. This enables the
derivation of more relevant and accurate estimates of parameters, such as the times of formation
and growth and formation rates, than has been possible before.
The results show the following features of NPF events in Beijing:
•   NPF events occur during clean air episodes when the wind direction is from the north of the

city.

•   We have provided the first temporal distribution chart of NPF events in Beijing which shows

that all but one of the 26 events began between 7:30 and 10:00 am.

•   The main characteristics of the particles in the NPF events are presented in Table 3.
•   In general, less than 10% of particles were charged and less than 1% of the clusters were

charged.

•   The fraction of particles that are charged was normally about 15%. This fraction increased to

20-30% during haze events and decreased to below 10% during NPF events.




**Acknowledgements**
This project was supported by the Australia-China Centre for Air Quality Science and Management.



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





**Figures**

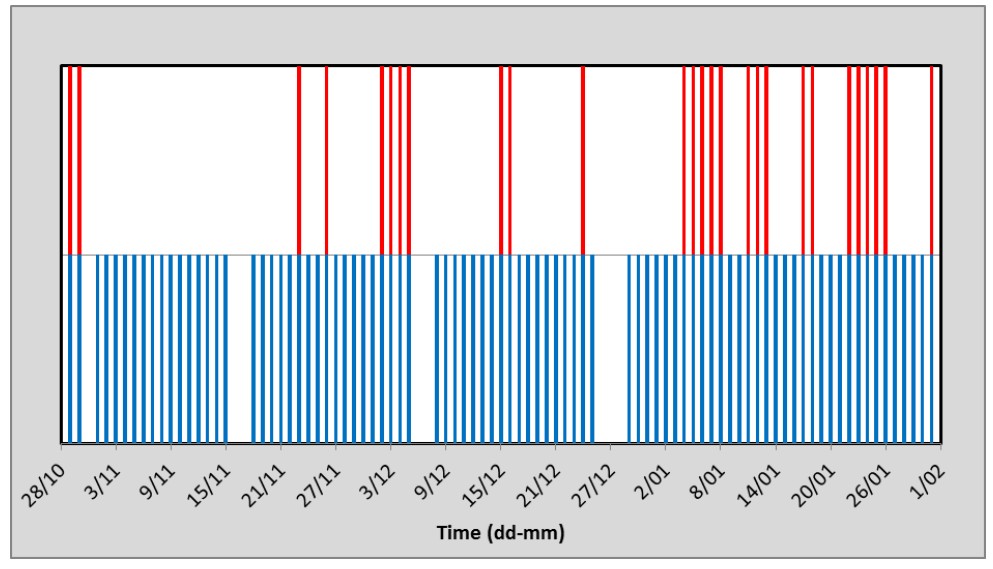


Figure 1: Summary of observational days (lower panel in blue) and days with NPF events (upper

panel in red).













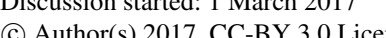




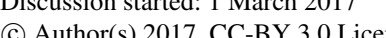

Figure 2: Daily values for January 2016: (a) NAIS Spectragram of PNC on a particle size–time diagram

(b) mean PM$_{2.5}$ concentration from the TEOM and (c) mean PNC in the size range 1.8 – 42

550            nm from the NAIS. The red and blue bars represent the NPF and Non-NPF days, respectively.



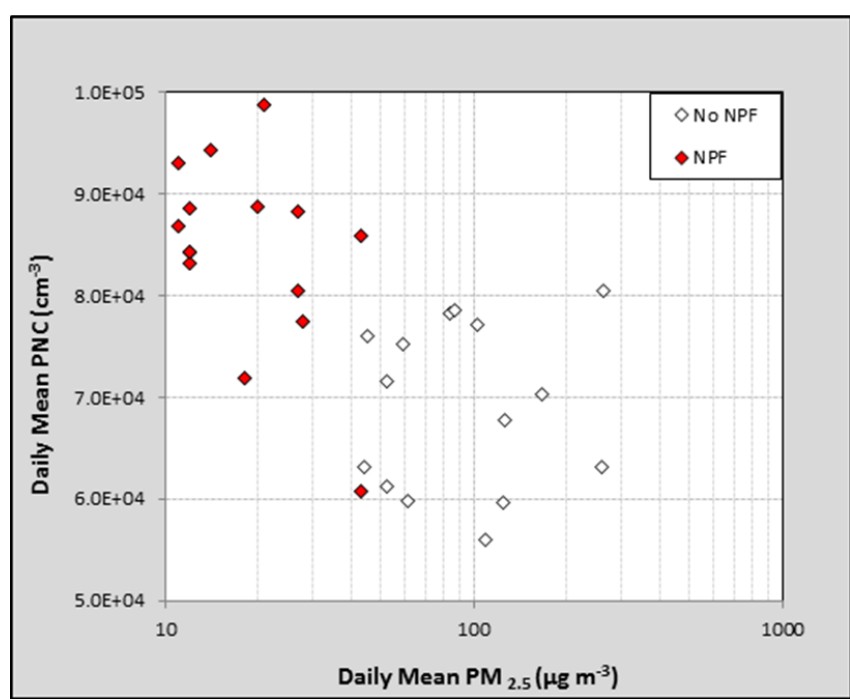



Figure 3: Daily mean PNC vs PM$_{2.5}$ for NPF days (filled markers) and no-NPF days (open markers)

during January 2016.




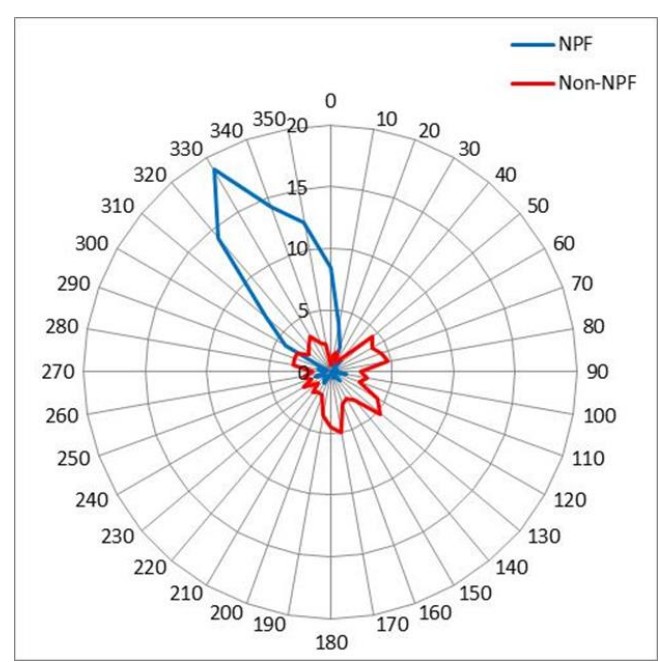



Figure 4: The wind direction rose for NPF days and non-NPF days during January. The radial scale

indicates percentages of time.


















Figure 5: Time series of total and charged (a) particles and (b) clusters during the period 30 Nov to 3

Dec as measured by the NAIS. 30 Nov and 1 Dec were haze days while two NPF events

occurred on 2 and 3 Dec.






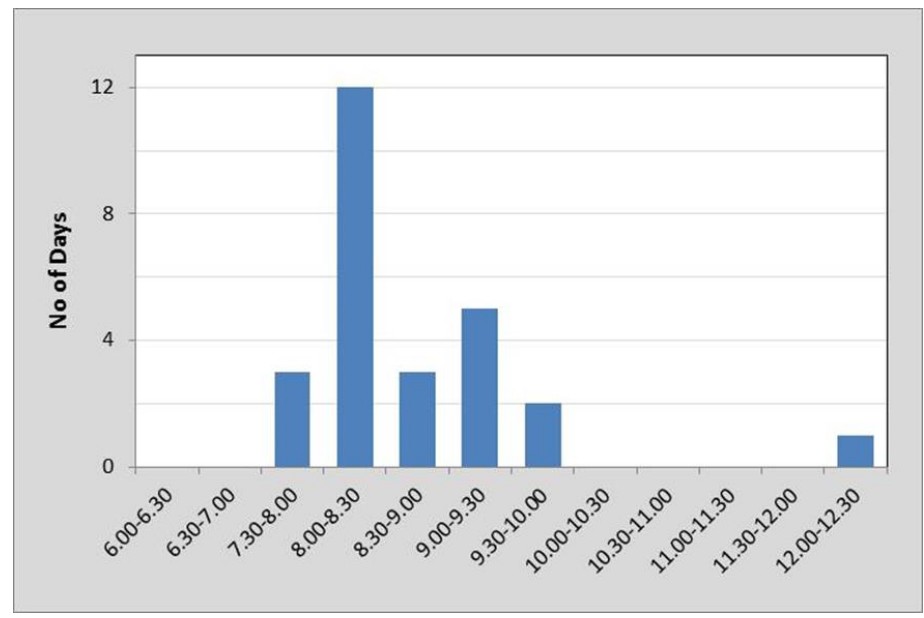


Figure 6: Distribution of the start times of the NPF events, classified into 30 min bins.











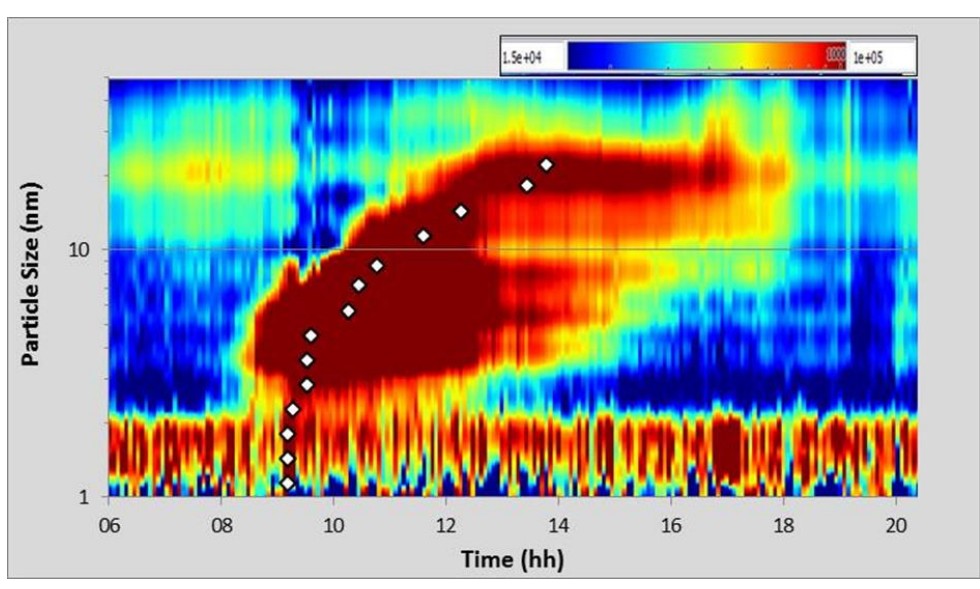


Figure 7: NAIS spectragram of the NPF event that occurred on 29[th] October. The clear banana shape

indicates strong particle growth. The markers show the median particle size at each time.

















**Tables**
Table 1: Summary of the observational days.

| Month | Total Days | Data Available Days | NPF Days : dN/dt >15000 cm$^{-3}$ h$^{-1}$ |
|---|---|---|---|
| October (28-31) | 4 | 2 | 2 |
| November (1-30) | 30 | 28 | 2 |
| December (1-31) | 31 | 26 | 6 |
| January (1-31) | 31 | 31 | 16 |
| Total | 96 | 87 | 26 |





















Table 2: Mean and peak values of neutral and charged particle and cluster concentrations during the
various types of days and events. The associated uncertainties in the values are up to 20%. The two
% columns show the respective charged/total percentages.

| | Particles (cm$^{-3}$) | | | Clusters (cm$^{-3}$) | | |
|---|---|---|---|---|---|---|
| | Neutral (x10$^4$) | Charged (x10$^4$) | % | Neutral (x10$^4$) | Charged (x10$^2$) | % |
| Normal Days (mean) | 5.9 | 1.0 | 15.0 | 3.1 | 1.5 | 0.5 |
| NPF Days (mean) | 8.0 | 0.9 | 10.1 | 2.6 | 1.4 | 0.5 |
| NPF Events (peak) | 23.7 | 1.4 | 5.4 | 4.9 | 3.3 | 0.7 |
| Haze Days (mean) | 5.0 | 2.0 | 28.2 | 3.8 | 2.4 | 0.6 |
| Haze Events (peak) | 12.3 | 3.1 | 20.0 | 9.9 | 4.8 | 0.5 |


















Table 3: Summary of mean and range of parameters calculated for the NPF events observed.

| Parameter | Mean | Range |
|---|---|---|
| Starting Time of NPF | 8.45 am | 7.30 am – 12.30 pm |
| Condensation sink ($s^{-1}$) | $5 \times 10^{-3}$ | $(2.1 – 8.9) \times 10^{-3}$ |
| Coagulation sink ($s^{-1}$) | $9 \times 10^{-4}$ | $(3.6 - 15.3) \times 10^{-4}$ |
| Formation rate ($J_2$) ($cm^{-3}\ s^{-1}$) | 23 | 10 - 36 |
| Growth rate ($nm\ h^{-1}$) | 3.5 | 0.5 - 9.0 |

