# Peer review of "Observations of Particles at their Formation Sizes in Beijing, China"

_Atmospheric Chemistry and Physics, 2017_

## Referee Comment (RC1) · Anonymous Referee #1 · 16 Mar 2017

This paper contributes to the understanding of some of the factors that control new particle formation (NPF) events in more-polluted regions of the atmosphere. The paper is well written and falls within the scope of the journal. I find the comparisons between NPF and non-NPF days to be of particular interest, along with the detailed measurements of NPF events in the ~2-10 nm range by the NAIS, a range not well-captured by studies that rely on SMPS-type particle number concentration measurements alone. I recommend this paper to be published in ACP with minor revisions, as discussed below.

General comments:

Page 4, lines 81-83: This statement is confusing. Did the PNC exceed 105 cm-3 on all 45 days or just for the 25 days that NPF was observed? Please clarify.

[Figure]

Page 8, calculation of the diffusion coefficient: It would be good to include a brief discussion of the assumption that the main condensing vapor is sulfuric acid. Particle composition measurements were not a part of this work, but the authors do cite Yue et al. (2010) as showing that some NPF events in Beijing had sulfuric acid accounting for much less than half of the total growth rate, with organics accounting for ∼55% of the growth. In the kinetic regime, the RMS speed of a molecule depends on 1/sqrt(MW), where MW = molecular weight = 98 g/mol for sulfuric acid = ∼200 g/mol for organics. This would mean that organics would be about ∼30% slower, and condensation in the kinetic regime is proportional to RMS speed. The continuum regime is trickier as it depends on the diffusion coefficient instead of RMS speed, but if we simplify to assume everything is in the kinetic regime, then the CS would scale as sqrt(MW of sulfuric acid) / sqrt (MW of orgs) ∼ sqrt(98) / sqrt(200). There are of course limited calculations and measurements of the diffusion coefficients of organic molecules as a function of temperature; however the authors could briefly comment on some of the literature values compared to their assumed value of D using sulfuric acid.

Page 9, condensation sink (CS) calculations: Why did the authors (1) choose to use 303 K in their diffusion coefficient (D) calculation (line 204) and (2) only use the SMPS PNC for the CS (line 203)? In regards to (1): temperature data wasn't reported in this paper but was taken as part of the meteorological data. The historical data reports Beijing's average temperature in January as being around ∼270 K. This difference in temperature doesn't lead to a particularly large change in D but certainly is worth addressing. In regards to (2): I would like to know why the authors chose to neglect the PNC data obtained from the NAIS for <14 nm size bins. A few calculations with "toy" size distributions show that, depending on the number concentration at these <14 nm bins, the CS can be non-trivially changed with the inclusion of these smaller bins. If the size distributions during NPF events in this paper such that the CS hardly changes with the inclusion of the smaller size bins, this should be stated. Page 13, lines 294-296: It is also worth mentioning that the pre-existing particles coming into the region from the winds from the south are also increasing the condensation sink, further reducing the

likelihood of NPF.

Figures/Tables:

Each figure (excepting Fig 4) could benefit from being more professionally presented. I'm not sure what programming language was used to create these figures but if it is e.g. python, using savefig('name.png',dpi=300) and savefig('name.pdf') would create much nicer looking figures. The text is somewhat blurry and could benefit from being saved at a higher dpi (for png) or as a pdf without the grey backgrounds.

Figures 2 and 7: The colorbars need labels of units. The numbers on the colorbars are quite blurry and need to be sharpened.

Technical comments:

Abstract, lines 29-31: The sentence would read better is if said 'Estimated characteristics...are very different than to when the measurements. . ..

Page 3, line 57: environments (needs an 's')

Page 11, line 247: no comma after 'that'.

Page 16, line 385: This sentence might read better if it said '...in the smallest particle size bin 2-3 nm for the times at which the rate of increase. . .'

---

## Referee Comment (RC2) · Anonymous Referee #2 · 29 Mar 2017

**General comments:**

For this paper, the authors employed a Neutral cluster and Air Ion Spectrometer (NAIS) to investigate the early steps of new-particle formation (NPF) events in Beijing, China, over a period of 3 months. Specifically, observations were made down to particle (or cluster) sizes of about 2 nm. NPF events in large, polluted urban areas, in particular in E Asia, are a current subject of atmospheric research (e.g. Kulmala et al., 2017). To my knowledge, this is the first report on deploying an NAIS in a Chinese megacity for this purpose, and it constitutes one of recent attempts of improving on the observations of NPF in such environments by directly measuring in the sub-3 nm size range (cf. Cai and Jiang, 2017; Yu et al., 2016). As such, the study is timely and of interest to the scientific community engaged in this field, and I recommend its publication in Atmospheric Chemistry and Physics. Before that however, I recommend a major revision to take care of some important issues.

My main concern with the study in its present form is the treatment and discussion of the NAIS measurements for the sub-3 nm size range. The treatment, presentation and interpretation of these data need to be brought into a form more rigorously consistent within the paper itself, as well as with best-practices recommended by the community (Manninen et al., 2016) – in particular as the corresponding results are a major selling point here.

**Comments regarding sub-3 nm measurements:**

Lines 109-110: "The NAIS … can detect particles down to a size of 0.8 nm":
My main point is that the NAIS can actually *not* be used to measure *neutral* compounds down to this size, so this statement is misleading in its current form. The NAIS does detect ions with the corresponding mobility, but due to the interference from charger ions it is deemed not possible to determine concentrations of neutral clusters for the smallest size bins. Quoting Manninen et al. (2016), which is cited also in this paper (line 137), "the particles below about 2 nm cannot be reliably distinguished from the corona-generated ions. Typically, the lowest detection limit for the NAIS in the particle mode is between 2 and 3 nm depending on the corona voltage and on the properties and composition of carrier gas (environmental conditions)." Details can be found in their paper and references therein.

At one occasion later, the authors appear to consider this instrumental limitation, e.g. section 2.2.3.

Then, section 3.4 (including Fig. 5 and Table 2) discusses charged vs. neutral "cluster" and "particle" concentrations. Here, the authors need to state what is their definition of "cluster" and "particle". And in light of the above, they might need to reconsider if total neutral cluster concentrations (as implied in section 3.4) can even be derived from the NAIS measurements! The discussions throughout section 3.4 may have to be revised…
E.g., depending on those definitions, could the observed decreases of "neutral clusters" for NPF days (e.g. Fig. 5b) be explained by instrument response to a change in environmental conditions?

Figure 2, top panel, and Figure 7:
As a consequence, I would argue that particle size distribution data below 2 nm shouldn't even be shown. The concentrations at the size bins <2 nm are subject to instrumental factors, not necessarily resulting from actual variations in the concentrations of sub-2 nm neutral clusters (particles). Hence, their display here could prompt an unaware reader to draw wrong conclusions about the actual population of sub-2 nm neutral clusters.

**Other comments:**

Line 68: Kulmama should probably be Kulmala – also in later instances for this reference. Speaking of which, the recent paper by Kulmala et al. (2017) is relevant to this study and should be brought to attention in the introduction.
As condensation sinks were calculated for this study, it might be useful even to shortly discuss the authors' findings in light of the conclusions of that paper (see e.g. lines 237-239).

Also, it could be interesting to compare the results here with those in Yu et al. (2016). Therein, they report in particle formation and growth rates during NPF events in Nanjing, also down to sub-3nm sizes.

Lines 277 & Fig. 3, line 287:
"Haze days" seems to be used interchangeably with "no-NPF days". Are they? If so, that point should be made clearer. If not, it may be feasible to mark them in Fig. 3.
The various types of day are actually defined later on (lines 325-329). I suggest moving this definition to an earlier place, and then shortly mention it again later.

Line 318: "attachment to existing particles"
I would have expected this process be more pronounced on the *no*-NPF days, when condensation sinks were higher.

Line 378: "previous have not been able"
I assume the authors refer to their novel measurement of particles in the 2-3 nm allowing them to more accurately calculating the coagulation sink (CoagS) for particles down to 2 nm. That's technically OK, but one would expect those small particles (i.e. in the 2-3 nm range for instance) to play a minor (negligible?) role in determining CoagS. How much is the value obtained here improved (increased) by the possibility to take the 2-3 nm range into account?

**Minor comments:**

Abstract, 2nd sentence: The statement should be clarified. From what are the estimated characteristics different in the case of restricted measurements?

Lines 152-153: It may be interesting and instructive for the reader to hear, in short, about the nature of the problems encountered.

Line 263: Does this t-test result apply to the whole measurement campaign, or just the subset shown in Fig. 2? In the latter case, would it change when applied to the whole period?

Line 297: "are more likely …" than what else?

Lines 329-332, Table 2: The source of the uncertainty of 20% has remained unclear to me. Maybe the authors can rephrase.

Most figures have a gray background and odd dark-gray or blank frames. They would look better without any that.

The text/numbers in the color bar in Figures 2 and 7 are difficult to read and lack units.

**References:**

Cai, R. and Jiang, J.: A new balance formula to estimate new particle formation rate: reevaluating the effect of coagulation scavenging, Atmos. Chem. Phys. Discuss., in review, doi:10.5194/acp-2017-199, 2017.

Kulmala, M., Kerminen, V.-M., Petäjä, T., Ding, A. and Wang, L.: Atmospheric Gas-to-Particle Conversion: why NPF events are observed in megacities?, Faraday Discuss., doi:10.1039/C6FD00257A, 2017.

Manninen, H. E., Mirme, S., Mirme, A., Pet??j??, T. and Kulmala, M.: How to reliably detect molecular clusters and nucleation mode particles with Neutral cluster and Air Ion Spectrometer (NAIS), Atmos. Meas. Tech., 9(8), 3577–3605, doi:10.5194/amt-9-3577-2016, 2016.

Yu, H., Zhou, L., Dai, L., Shen, W., Dai, W., Zheng, J., Ma, Y. and Chen, M.: Nucleation and growth of sub-3 nm particles in the polluted urban atmosphere of a megacity in China, Atmos. Chem. Phys., 16(4), 2641–2657, doi:10.5194/acp-16-2641-2016, 2016.

---

## Author Comment (AC1) · 24 Apr 2017

**Response to Comments from Anonymous Referee #1**

**Overall Comments**

This paper contributes to the understanding of some of the factors that control new particle formation (NPF) events in more-polluted regions of the atmosphere. The paper is well written and falls within the scope of the journal. I find the comparisons between NPF and non-NPF days to be of particular interest, along with the detailed measurements of NPF events in the ~2-10 nm range by the NAIS, a range not well-captured by studies that rely on SMPS-type particle number concentration measurements alone. I recommend this paper to be published in ACP with minor revisions, as discussed below.

**Response to Overall Comments**

We thank the reviewer for these positive comments and are glad to note that the paper falls within the scope of the journal.

**General comments:**

**Comment 1**

Page 4, lines 81-83: This statement is confusing. Did the PNC exceed $10^5$ cm$^{-3}$ on all 45 days or just for the 25 days that NPF was observed? Please clarify.

**Response 1**

We accept that this sentence is confusing. We will amend it as follows:

*"They observed NPF on 25 out of 45 days of measurement, and on each of these days the PNC exceeded $10^5$ cm$^{-3}$."*

**Comment 2**

Page 8, calculation of the diffusion coefficient: It would be good to include a brief discussion of the assumption that the main condensing vapor is sulfuric acid. Particle composition measurements were not a part of this work, but the authors do cite Yue et al. (2010) as showing that some NPF events in Beijing had sulfuric acid accounting for much less than half of the total growth rate, with organics accounting for ~55% of the growth. In the kinetic regime, the RMS speed of a molecule depends on 1/sqrt(MW), where MW = molecular weight = 98 g/mol for sulfuric acid = ~200 g/mol for organics. This would mean that organics would be about ~30% slower, and condensation in the kinetic regime is proportional to RMS speed. The continuum regime is trickier as it depends on the diffusion coefficient instead of

RMS speed, but if we simplify to assume everything is in the kinetic regime, then the CS would scale as sqrt(MW of sulfuric acid) / sqrt (MW of orgs) ~ sqrt(98) / sqrt(200). There are of course limited calculations and measurements of the diffusion coefficients of organic molecules as a function of temperature; however the authors could briefly comment on some of the literature values compared to their assumed value of D using sulfuric acid.

**Response 2**

In response to these comments, we will incorporate some or all of the following discussion into the paper:

It is now well established that sulfuric acid is the key precursor gas in nucleation, although low vapour pressure organics may contribute to the subsequent aerosol growth (Curtius, 2006). Sulfuric acid has a low vapour pressure which is reduced further in the presence of water. When produced from $SO_2$ in the gas phase, it is easily supersaturated and begins to condense. Moreover, most of the particles in the atmosphere are in the kinetic regime (smaller than 0.01 µm)(Seinfeld and Pandis, 2006). In this regime, condensation is directly proportional to the RMS speed of the molecules. The RMS speed is inversely proportional to the square root of the molecular weight of the molecule. Thus, a sulfuric acid molecule, with a molecular weight of 98 g mol$^{-1}$, has an RMS speed that is about 30% higher than a typical organic gas molecule with a molecular weight of about 200 g mol$^{-1}$, Thus, condensation of sulfuric acid will occur much more readily than organic molecules. Studies in Beijing have confirmed that NPF is more likely to occur in a sulfur-rich environment than in one that is sulfur-poor ((Yue et al., 2010;Guo et al., 2014;Wu et al., 2007)). Wu et al. (2007) also assumed that sulfuric acid was the main condensable vapour in determining the particle formation rates during NPF events in Beijing.

Our estimated values of D for sulfuric acid using the equation given in Jeong (2009) are 0.092 cm$^2$ s$^{-1}$ at 303K and 0.087 cm$^2$ s$^{-1}$ at 273K. The value of 0.092 cm$^2$ s$^{-1}$ at 303K is reasonable as it is similar to other values given in the literature at room temperature, for example Brus et al. (2016) (0.08 cm$^2$ s$^{-1}$ ) and Eisele and Hanson (2000) (0.095 cm$^2$ s$^{-1}$).
The values of D for common organic trace gases as given in the literature are somewhat smaller than this, e.g. 0.07 cm$^2$ s$^{-1}$ for isoprene (Tang et al., 2015) and terpenes (Williams, 2004). D for atmospheric amines are of the same order as that for sulfuric acid (Lugg, 1968) Therefore, we feel that D = 0.092 cm$^2$ s$^{-1}$ is a reasonable value to use in calculating the CS.

**Comment 3**

Page 9, condensation sink (CS) calculations: Why did the authors (1) choose to use 303 K in their diffusion coefficient (D) calculation (line 204) and (2) only use the SMPS PNC for the CS (line 203)? In regards to (1): temperature data wasn't reported in this paper but was taken as part of the meteorological data. The historical data reports Beijing's average temperature in January as being around ~270 K. This difference in temperature doesn't lead to a particularly large change in D but certainly is worth addressing.

**Response 3**

3 (1) We accept the point about the temperature. The average temperature in Beijing during the observations was close to 273K. We have re-calculated our parameters using this value for T. The revised values are given below, and these will replace the values in the paper.

3 (2) Regarding the point about calculating the CS, our response to this comment is included in Response 4 below.

**Comment 4**

In regards to (2): I would like to know why the authors chose to neglect the PNC data obtained from the NAIS for <14 nm size bins. A few calculations with "toy" size distributions show that, depending on the number concentration at these <14 nm bins, the CS can be non-trivially changed with the inclusion of these smaller bins. If the size distributions during NPF events in this paper such that the CS hardly changes with the inclusion of the smaller size bins, this should be stated.

**Response 4**

We re-calculated the parameters, first by holding the temperature at 303K, in order to check if including the particles smaller than 14 nm will make a significant difference to the CS. We found that the CS value increased by about 8% (from 4.8 to 5.2 $s^{-1}$).

Therefore, we have re-calculated all the parameters using the value of CS obtained across the entire size range (< 14 nm from the NAIS plus >14 nm from the SMPS) and with the temperature changed from 303K to 273K.

The original values in the paper are as follows:

D = 0.092 $cm^2$ $s^{-1}$
CS = 4.8x$10^{-3}$ $s^{-1}$
Coag = 8.3x$10^{-4}$ $s^{-1}$
FR = 23 $cm^{-3}$ $s^{-1}$

The new values obtained are as follows:

D = 0.087 $cm^2$ $s^{-1}$
CS = 4.2x$10^{-3}$ $s^{-1}$
Coag = 7.2x$10^{-4}$ $s^{-1}$
FR = 26 $cm^{-3}$ $s^{-1}$

We will correct the paper accordingly.

**Comment 5**

Page 13, lines 294-296: It is also worth mentioning that the pre-existing particles coming into the region from the winds from the south are also increasing the condensation sink, further reducing the likelihood of NPF.

**Response 5**

We will include the following text.

*"Pre-existing particles entering the region with the winds from the south will also increase the condensation sink, further reducing the likelihood of NPF."*

**Figures/Tables:**

**Comment 6**

Each figure (excepting Fig 4) could benefit from being more professionally presented. I'm not sure what programming language was used to create these figures but if it is e.g. python, using savefig('name.png',dpi=300) and savefig('name.pdf') would create much nicer looking figures. The text is somewhat blurry and could benefit from being saved at a higher dpi (for png) or as a pdf without the grey backgrounds.

**Response 6**

Since first submission to ACP required embedding the figures within the body of the manuscript, the resolution of the figures has suffered. In the final submission, we shall present each figure as a separate file, so that they will be of much higher resolution. Also, we will remove the grey background and the frames from all the figures.

**Comment 7**

Figures 2 and 7: The colorbars need labels of units. The numbers on the colorbars are quite blurry and need to be sharpened.

**Response 7**

The numbers that appear on the color bars are produced by the software. As we have provided two labels with the end point values, we feel that these intermediate numbers on the color bars are not essential. To compensate for this, we will further sharpen the text labels at the two ends of the color bars.
We will add the units ($cm^{-3}$).

**Technical comments:**

**Comment 8**

Abstract, lines 29-31: The sentence would read better is if said 'Estimated characteristics... are very different than to when the measurements. . ..

**Response 8**

The text will be changed as requested.

**Comment 9**

Page 3, line 57: environments (needs an 's')

**Response 9**

The "s" will be added.

**Comment 10**

Page 11, line 247: no comma after 'that'.

**Response 10**

The comma will be deleted.

**Comment 11**

Page 16, line 385: This sentence might read better if it said '...in the smallest particle size bin 2-3 nm for the times at which the rate of increase. . .'

**Response 11**

We agree. The text will be changed as requested.

References:

Brus, D., Skrabalova, L., Herrmann, E., Olenius, T., Travnickova, T., and Merikanto, J.: Temperature-dependent diffusion coefficient of H2SO4 in air: laboratory measurements using laminar flow technique, Atmos. Chem. Phys. Discuss., 2016, 1-26, 10.5194/acp-2016-398, 2016.
Curtius, J.: Nucleation of atmospheric aerosol particles, Comptes Rendus Physique, 7, 1027-1045, 2006.
Eisele, F., and Hanson, D.: First measurement of prenucleation molecular clusters, The Journal of Physical Chemistry A, 104, 830-836, 2000.
Guo, S., Hu, M., Zamora, M. L., Peng, J., Shang, D., Zheng, J., Du, Z., Wu, Z., Shao, M., and Zeng, L.: Elucidating severe urban haze formation in China, Proceedings of the National Academy of Sciences, 111, 17373-17378, 2014.
Jeong, K.: Condensation of water vapor and sulfuric acid in boiler flue gas, ProQuest, 2009.
Lugg, G.: Diffusion coefficients of some organic and other vapors in air, Analytical Chemistry, 40, 1072-1077, 1968.
Tang, M. J., Shiraiwa, M., Pöschl, U., Cox, R. A., and Kalberer, M.: Compilation and evaluation of gas phase diffusion coefficients of reactive trace gases in the atmosphere: Volume 2. Diffusivities of organic compounds, pressure-normalised mean free paths, and average Knudsen numbers for gas uptake calculations, Atmos. Chem. Phys., 15, 5585-5598, 10.5194/acp-15-5585-2015, 2015.
Williams, J.: Organic trace gases in the atmosphere: an overview, Environ. Chem., 1, 125-136, 2004.
Wu, Z., Hu, M., Liu, S., Wehner, B., Bauer, S., Wiedensohler, A., Petäjä, T., Dal Maso, M., and Kulmala, M.: New particle formation in Beijing, China: Statistical analysis of a 1-year data set, Journal of Geophysical Research: Atmospheres, 112, 2007.
Yue, D., Hu, M., Zhang, R., Wang, Z., Zheng, J., Wu, Z., Wiedensohler, A., He, L., Huang, X., and Zhu, T.: The roles of sulfuric acid in new particle formation and growth in the mega-city of Beijing, Atmospheric Chemistry and Physics, 10, 4953-4960, 2010.

---

## Author Comment (AC2) · 24 Apr 2017

**Response to Comments from Anonymous Referee #2**

**General comments:**

For this paper, the authors employed a Neutral cluster and Air Ion Spectrometer (NAIS) to investigate the early steps of new-particle formation (NPF) events in Beijing, China, over a period of 3 months. Specifically, observations were made down to particle (or cluster) sizes of about 2nm. NPF events in large, polluted urban areas, in particular in E Asia, are a current subject of atmospheric research (e.g. Kulmala et al., 2017). To my knowledge, this is the first report on deploying an NAIS in a Chinese megacity for this purpose, and it constitutes one of recent attempts of improving on the observations of NPF in such environments by directly measuring in the sub-3 nm size range (cf. Cai and Jiang, 2017; Yu et al., 2016). As such, the study is timely and of interest to the scientific community engaged in this field, and I recommend its publication in Atmospheric Chemistry and Physics.

**Response to General Comments**

We thank the reviewer for these positive comments and are glad to note that he/she feels that the paper would be of interest to the scientific community engaged in this field, and for recommending that, subject to the changes below, it is suitable for publication in Atmospheric Chemistry and Physics.

**Major Comments 1**

Before that however, I recommend a major revision to take care of some important issues.

My main concern with the study in its present form is the treatment and discussion of the NAIS measurements for the sub-3 nm size range. The treatment, presentation and interpretation of these data need to be brought into a form more rigorously consistent within the paper itself, as well as with best-practices recommended by the community (Manninen et al., 2016) – in particular as the corresponding results are a major selling point here.

**Comments regarding sub-3 nm measurements:**
Lines 109-110: "The NAIS … can detect particles down to a size of 0.8 nm":
My main point is that the NAIS can actually *not* be used to measure *neutral* compounds down to this size, so this statement is misleading in its current form. The NAIS does detect ions with the corresponding mobility, but due to the interference from charger ions it is deemed not possible to determine concentrations of neutral clusters for the smallest size bins. Quoting Manninen et al. (2016), which is cited also in this paper (line 137), "the particles below about 2 nm cannot be reliably distinguished from the corona-generated ions. Typically, the lowest detection limit for the NAIS in the particle mode is between 2 and 3 nm depending on the corona voltage and on the properties and composition of carrier gas (environmental conditions)." Details can be found in their paper and references therein. At one occasion later, the authors appear to consider this instrumental limitation, e.g. section 2.2.3.

**Response to Major Comments 1**

We agree with these comments and accept that the NAIS has a problem in differentiating between charged and neutral particles and clusters at sizes below 2.0 nm owing to the presence of corona-generated ions as pointed out by Asmi et al. (2009), Manninen et al. (2011) and Manninen et al. (2016).

We will address this issue by making the following changes to the paper:

1. Considering the limitations of the NAIS in measuring total particle and cluster concentrations at sizes smaller than 2 nm, we will restrict our observations to particles that are larger than 2.0 nm.
2. This will result in the smallest size bin (1.6-2.0 nm) being excluded from the particle analyses.
3. We estimate that this will decrease our charged and neutral PNC values by about 5%. We will make this change right through the manuscript.
4. In Table 2, we will remove the two columns showing charged and neutral cluster concentrations.
5. In Figure 7, we will remove the three points below 2.0 nm and insert a note in the caption cautioning against using the data below 2.0 nm.
6. Similarly, in Figure 2(a), we will insert a note stating that the data below 2.0 nm should be treated with caution.

In addition, we will incorporate the following changes to the text:

Lines 109-110: we will remove the text "can detect particles down to a size of 0.8 nm" and amend the sentence as follows:

*"The NAIS is specifically designed to monitor particle formation as it can detect clusters and particles down to a size of 0.8 nm"*

Line 136 - : We will insert the following text:

*"However, Asmi et al. (2009), Manninen et al. (2011) and Manninen et al. (2016) have pointed out that the lowest detection limit for the NAIS in the particle mode is about 2.0 nm owing to the presence of corona-generated ions. Therefore, at sizes smaller than 2.0 nm, the NAIS cannot reliably distinguish between charged and neutral particles Therefore, Manninen et al. (2011) specified the lowest detection limit of the NAIS to be 1.6 and 1.7 nm for negative and positive ions, respectively, and 2.0 nm for neutral particles. Therefore, in this study, we will restrict our observations to the particle size range 2.0-42 nm."*

Section 2.1.1 Line 165 - : we will amend the text as follows:

*"..where N is the number of particles in the size range 2.0 -10.0 nm."*

While changing the definition of the lower end of N from 1.8 nm to 2.0 nm affected the total PNC in that size range by about 5%, it did not affect the decisions regarding the identification of any of the NPF events.

Line 170: The text will be changed to

*"The starting times of an event was determined by using the time of sudden increase in total PNC in the size range 2.0 – 10.0 nm."*

Line 223: The text will be changed to

*"…we exported the number concentrations of particles obtained from the NAIS in 14 bins in the size range 2.0 – 42.0 nm."*

**Major Comments 2**

Section 3.4 (including Fig. 5 and Table 2) discusses charged vs. neutral "cluster" and "particle" concentrations. Here, the authors need to state what is their definition of "cluster" and "particle". And in light of the above, they might need to reconsider if total neutral cluster concentrations (as implied in section 3.4) can even be derived from the NAIS measurements! The discussions throughout section 3.4 may have to be revised. E.g., depending on those definitions, could the observed decreases of "neutral clusters" for NPF days (e.g. Fig. 5b) be explained by instrument response to a change in environmental conditions?

**Response to Major Comments 2**

We will provide the conventional definitions of clusters and particles from the literature in our introduction. In Section 3.4, we will restrict our analysis to particles larger than 2.0 nm. This will result in the smallest size bin (1.6-2.0 nm) being excluded from the particle analyses. We estimate that this will decrease our charged and neutral PNC values by about 5%. We will make this change right through the manuscript. The sub-heading title will be changed to "*Charged Particles*". All references to cluster concentrations will be removed. In Fig 5, we will remove Fig 5(b) that shows the neutral and charged cluster concentrations. Fig 5 will now show only the neutral and charged particle concentrations.

In Table 2, we will remove the two columns showing charged and neutral cluster concentrations.

**Major Comments 3**

Figure 2, top panel, and Figure 7:
As a consequence, I would argue that particle size distribution data below 2 nm shouldn't even be shown. The concentrations at the size bins <2 nm are subject to instrumental factors, not necessarily resulting from actual variations in the concentrations of sub-2 nm neutral clusters (particles). Hence, their display here could prompt an unaware reader to draw wrong conclusions about the actual population of sub-2 nm neutral clusters.

**Response to Major Comments 2**

In Figure 7, we will remove the three points below 2.0 nm and insert a note in the caption cautioning against using the data below 2.0 nm.

Similarly, in Figure 2(a), we will insert a note stating that the data below 2.0 nm should be treated with caution.

**Other comments:**

**Comment 1**

Line 68: Kulmama should probably be Kulmala – also in later instances for this reference.

**Response 1**

In the journal paper the name has been mis-spelt as 'Kulmama". Correcting this is bound to affect the citation count and we will seek the advice of the Editor on this matter and make the change, if required.

**Comment 2**

Speaking of which, the recent paper by Kulmala et al. (2017) is relevant to this study and should be brought to attention in the introduction. As condensation sinks were calculated for this study, it might be useful even to shortly discuss the authors' findings in light of the conclusions of that paper (see e.g. lines 237-239).

**Response 2**

We agree. We will include the following text in the Introduction:

*"A recent paper by Kulmala et al. (2017) attempts to explain the occurrence of NPF under highly polluted conditions by combining direct observations and conceptual modelling. They hypothesized that the apparent discrepancy may be explained if the coagulation rate of molecular clusters by particles is significantly less than their collision rate or if the clusters grew much faster than is normally expected".*

In the discussion section, we will also consider the dimensionless parameter, p, defined by Kulmala et al. (2017) and compare the value estimated from our results in the present study with the value predicted by Kulmala et al. (2017) for a polluted city like Beijing.

**Comment 3**

Also, it could be interesting to compare the results here with those in Yu et al. (2016). Therein, they report in particle formation and growth rates during NPF events in Nanjing, also down to sub-3nm sizes.

**Response 3**

We agree. We will include the formation rate and growth rate values found by Yu et al. (2016) in Nanjing by inserting the following text into the discussion section:

*"These values may be compared with that found by Yu et al. (2016) in the urban atmosphere of Nanjing, China. They studied eight NPF events using a nano-condensation nucleus counter system capable of measuring particle size distributions down to 1.4 nm and estimated initial*

*and peak particle formation rates of 210 and 2500 cm$^{-3}$ s$^{-1}$, respectively. The formation rates showed good linear correlation with a sulfuric acid proxy".*

*"Yu et al. (2016) determined a relatively high local maximum growth rate of 25 nm h$^{-1}$ in Nanjing, China.*

**Comment 4**

Lines 277 & Fig. 3, line 287:
"Haze days" seems to be used interchangeably with "no-NPF days". Are they? If so, that point should be made clearer. If not, it may be feasible to mark them in Fig. 3. The various types of day are actually defined later on (lines 325-329). I suggest moving this definition to an earlier place, and then shortly mention it again later.

**Response 4**

In Figure 3, the NPF days are shown as red full markers. The points shown in white hollow markers are all other days, including normal (no-NPF) and haze days. We will change the figure caption to read "other days" instead of "No-NPF Days".

Also, we will move the definition to the methods section and shortly mention it again at this point.

**Comment 5**

Line 318: "attachment to existing particles"

**Response 5**
This sentence will be removed when the discussion on cluster concentrations is excluded.

**Comment 6**

I would have expected this process be more pronounced on the *no*-NPF days, when condensation sinks were higher.

**Response 6**

This statement will also be removed when the discussion on cluster concentrations is excluded. However, we calculated the condensation sinks for no-NPF days and found that it was 0.006 s$^{-1}$, which is not significantly higher than the corresponding value on NPF days (0.005 s$^{-1}$). However, the condensation sink on haze days was 0.060 s$^{-1}$, which is significantly higher than both normal days and NPF days. We will insert the following text into the end of section 3.6 on condensation sinks:

*"The value of the condensation sink during NPF events (0.005 s$^{-1}$) was not significantly different to the corresponding average values during other times on NPF days and on normal days with no NPF (0.006 s$^{-1}$). However, the mean condensation sink on haze days (0.060 s$^{-1}$) was significantly higher than both these values."*

**Comment 7**

Line 378: "previous have not been able"
I assume the authors refer to their novel measurement of particles in the 2-3 nm allowing them to more accurately calculating the coagulation sink (CoagS) for particles down to 2 nm. That's technically OK, but one would expect those small particles (i.e. in the 2-3 nm range for instance) to play a minor (negligible?) role in determining CoagS. How much is the value obtained here improved (increased) by the possibility to take the 2-3 nm range into account?

**Response 7**

Equation (5) for the formation rate considers the particles in the size range 2-3 nm. The rate of change of the number of particles in this size range was not available to previous workers. We use this, together with the coagulation sink of the particles in the size range 2-3 nm ($CoagSd_p$) to calculate the formation rate. The coagulation rate CoagS refers to the entire particle size range and this value is, as the reviewer points out, much larger than $CoagSd_p$ in the size range 2-3 nm. However, we thank the reviewer for this comment as it shows that the text was not very clear. We will modify the text to make this as clear as possible.

**Minor comments:**

**Comment 8**

Abstract, 2nd sentence: The statement should be clarified. From what are the estimated characteristics different in the case of restricted measurements?

**Response 8**

We will re-word this sentence as follows:
*"Estimated characteristics of NPF events, such as their starting times and formation and growth rates of particles, are more accurate when the detection size of particles is smaller".*

**Comment 9**

Lines 152-153: It may be interesting and instructive for the reader to hear, in short, about the nature of the problems encountered.

**Response 9**

This sentence will be replaced by the following text:
*"Data was lost on nine days owing to various problems such as the loss of power, software malfunction and a blocked filter during a haze event."*

**Comment 10**

Line 263: Does this t-test result apply to the whole measurement campaign, or just the subset shown in Fig. 2? In the latter case, would it change when applied to the whole period?

**Response 10**

This was for the subset shown in Fig 2. However, when we consider the entire monitoring period, the corresponding difference was even more significant. We have added the following sentence:

*"The corresponding difference was even more significant when considering the entire monitoring period where the mean daily values of $PM_{2.5}$ on NPF days and the other days were 21 $\mu g\ m^{-3}$ and 143 $\mu g\ m^{-3}$, respectively."*

**Comment 11**

Line 297: "are more likely ..." than what else?

**Response 11**

We have amended the text as follows:
*"Thus, the observed haze events are unlikely to be caused by in-situ new particle formation and more likely to be due to particles carried by the wind into the city or being prevented from escaping due to temperature inversions in the atmosphere".*

**Comment 12**

Lines 329-332, Table 2: The source of the uncertainty of 20% has remained unclear to me. Maybe the authors can rephrase.

**Response 12**

We agree that the statement is unclear. We will replace it with:

*"The values shown are the means of the average $PM_{2.5}$ concentrations over all the 24-hour days. The daily mean values varied from day to day, especially on days with NPF events or haze events mainly due to the different durations of these events. We estimated the standard deviation about these mean values to be 20%".*

**Comment 13**

Most figures have a gray background and odd dark-gray or blank frames. They would look better without any that.

**Response 13**

We will remove the grey background and frames around all figures.

**Comment 14**

The text/numbers in the color bar in Figures 2 and 7 are difficult to read and lack units.

**Response 14**

We will improve the quality of the numbers on the color bars and include units ($cm^{-3}$).

References:

Asmi, E., Sipilä, M., Manninen, H., Vanhanen, J., Lehtipalo, K., Gagné, S., Neitola, K., Mirme, A., Mirme, S., and Tamm, E.: Results of the first air ion spectrometer calibration and intercomparison workshop, Atmospheric Chemistry and Physics, 9, 141-154, 2009.
Kulmala, M., Kerminen, V.-M., Petäjä, T., Aijun, D., and Wang, L.: Atmospheric Gas-to-Particle Conversion: why NPF events are observed in megacities?, Faraday Discussions, 2017.
Manninen, H., Franchin, A., Schobesberger, S., Hirsikko, A., Hakala, J., Skromulis, A., Kangasluoma, J., Ehn, M., Junninen, H., and Mirme, A.: Characterisation of corona-generated ions used in a Neutral cluster and Air Ion Spectrometer (NAIS), Atmospheric Measurement Techniques, 4, 2767, 2011.
Manninen, H. E., Mirme, S., Mirme, A., Petäjä, T., and Kulmala, M.: How to reliably detect molecular clusters and nucleation mode particles with Neutral cluster and Air Ion Spectrometer (NAIS), Atmos. Meas. Tech. Discuss, 2016.
Yu, H., Zhou, L., Dai, L., Shen, W., Dai, W., Zheng, J., Ma, Y., and Chen, M.: Nucleation and growth of sub-3 nm particles in the polluted urban atmosphere of a megacity in China, Atmospheric Chemistry and Physics, 16, 2641-2657, 2016.

---

## Author Response (AR1)

**Professor Lidia Morawska**
International Laboratory for Air Quality and Health
Queensland University of Technology
George Street, Brisbane QLD 4001 Australia
Email: l.morawska@qut.edu.au

11$^{th}$ May 2017

Natascha Töpfer
Copernicus Publications
Editorial Support
editorial@copernicus.org

Dear Natascha,

**Submission of Revised Manuscript Number: acp-2017-156**

**Title:** Observations of Particles at their Formation Sizes in Beijing, China

**Authors (names and email addresses):**

Dr. Rohan Jayaratne:  r.jayaratne@qut.edu.au
Ms. Buddhi Pushpawela:  buddhi.pushpawela@hdr.qut.edu.au
Dr. Congrong He: c.he@qut.edu.au
Dr. Jian Gao: gaojian@craes.org.cn
Ms Hui Li: huili@craes.org.cn
Prof. Lidia Morawska: l.morawska@qut.edu.au

As requested, we have considered the comments of the two anonymous reviewers in detail and revised the paper accordingly.

I am submitting the following documents:

(1) Revised Manuscript (2) Revised Manuscript with all changes indicated in Track Changes (3) Detailed responses to Anonymous Reviewer 1 and (4) Detailed responses to Anonymous Reviewer 2.

I hope you will find it acceptable for publication in ACP.

There are a few matters that I would like to bring to your notice for consideration:

1. We would like "Rohan Jayaratne" and "Buddhi Pushpawela" to be listed as Joint First Authors of this paper. We do understand that only the first name will be listed on the system.

2. If possible, we would like to add one more co-author: Fahe Chai. His contact details are as follows: Chinese Research Academy of Environmental Sciences, Beijing 100012, China. Email: chaifahe@craes.org.cn

3. If possible, we would like to add Jian Gao as a co-corresponding author of this paper. His contact details are as follows: Chinese Research Academy of Environmental Sciences, Beijing 100012, China. Email: gaojian@craes.org.cn

4. Referring to "Other Comment 1" by Anon Reviewer 2, we are aware that Prof Kulmala's name has been mis-spelt in the journal paper Kulmama et al (2016) FESE. Correcting this in our paper would deny their paper of a citation. We leave this to your discretion and would be happy to accept your decision.

Thank you.

Please contact me at the email address below, should you have any further queries.

Yours sincerely,

[Figure]

**Professor Lidia Morawska, PhD**

**Director**
International Laboratory for Air Quality and Health
WHO CC for Air Quality and Health

**Director - Australia**
Australia – China Centre for Air Quality Science and Management
Queensland University of Technology
Phone: +61 7 3138 2616
Fax: +61 7 3138 9079
E-mail: l.morawska@qut.edu.au

This paper contributes to the understanding of some of the factors that control new particle formation (NPF) events in more-polluted regions of the atmosphere. The paper is well written and falls within the scope of the journal. I find the comparisons between NPF and non-NPF days to be of particular interest, along with the detailed measurements of NPF events in the ~2-10 nm range by the NAIS, a range not well-captured by studies that rely on SMPS-type particle number concentration measurements alone. I recommend this paper to be published in ACP with minor revisions, as discussed below.

**Response to Overall Comments**

We thank the reviewer for these positive comments and are glad to note that the paper falls within the scope of the journal.

**General comments:**

**Comment 1**

Page 4, lines 81-83: This statement is confusing. Did the PNC exceed $10^5$ cm$^{-3}$ on all 45 days or just for the 25 days that NPF was observed? Please clarify.

**Response 1**

We accept that this sentence is confusing. We have amended it on page 4, lines 92-93 as follows:

*"They observed NPF on 25 out of 45 days of measurement, and on each of these days the PNC exceeded $10^5$ cm$^{-3}$."*

**Comment 2**

Page 8, calculation of the diffusion coefficient: It would be good to include a brief discussion of the assumption that the main condensing vapor is sulfuric acid. Particle composition measurements were not a part of this work, but the authors do cite Yue et al. (2010) as showing that some NPF events in Beijing had sulfuric acid accounting for much less than half of the total growth rate, with organics accounting for ~55% of the growth. In the kinetic regime, the RMS speed of a molecule depends on 1/sqrt(MW), where MW = molecular weight = 98 g/mol for sulfuric acid = ~200 g/mol for organics. This would mean that organics would be about ~30% slower, and condensation in the kinetic regime is proportional to RMS speed. The continuum regime is trickier as it depends on the diffusion coefficient instead of

RMS speed, but if we simplify to assume everything is in the kinetic regime, then the CS would scale as sqrt(MW of sulfuric acid) / sqrt (MW of orgs) ~ sqrt(98) / sqrt(200). There are of course limited calculations and measurements of the diffusion coefficients of organic molecules as a function of temperature; however the authors could briefly comment on some of the literature values compared to their assumed value of D using sulfuric acid.

**Response 2**

In response to these comments, we have inserted the following text into the paper on page 8 and 9:

*"It is now well established that sulfuric acid is the key precursor gas in nucleation, although low vapour pressure organics may contribute to the subsequent aerosol growth (Curtius, 2006). Sulfuric acid has a low vapour pressure which is reduced further in the presence of water. When produced from $SO_2$ in the gas phase, it is easily supersaturated and begins to condense. Moreover, most of the particles in the atmosphere are in the kinetic regime (smaller than 0.01 μm)(Seinfeld and Pandis, 2006). In this regime, condensation is directly proportional to the RMS speed of the molecules. The RMS speed is inversely proportional to the square root of the molecular weight of the molecule. Thus, a sulfuric acid molecule, with a molecular weight of 98 g $mol^{-1}$, has an RMS speed that is about 30% higher than a typical organic gas molecule with a molecular weight of about 200 g $mol^{-1}$, Thus, condensation of sulfuric acid will occur much more readily than organic molecules. Studies in Beijing have confirmed that NPF is more likely to occur in a sulfur-rich environment than in one that is sulfur-poor ((Yue et al., 2010;Guo et al., 2014;Wu et al., 2007)). Wu et al. (2007) also assumed that sulfuric acid was the main condensable vapour in determining the particle formation rates during NPF events in Beijing".*

Our estimated values of D for sulfuric acid using the equation given in Jeong (2009) are 0.092 $cm^2$ $s^{-1}$ at 303K and 0.087 $cm^2$ $s^{-1}$ at 273K. The value of 0.092 $cm^2$ $s^{-1}$ at 303K is reasonable as it is similar to other values given in the literature at room temperature, for example Brus et al. (2016) (0.08 $cm^2$ $s^{-1}$ ) and Eisele and Hanson (2000) (0.095 $cm^2$ $s^{-1}$).
The values of D for common organic trace gases as given in the literature are somewhat smaller than this, e.g. 0.07 $cm^2$ $s^{-1}$ for isoprene (Tang et al., 2015) and terpenes (Williams, 2004). D for atmospheric amines are of the same order as that for sulfuric acid (Lugg, 1968) Therefore, we feel that D = 0.092 $cm^2$ $s^{-1}$ is a reasonable value to use in calculating the CS.

We have modified the text on page 10, lines 237-240 as follows:

*"The mean temperature in Beijing during the period of observation was close to 0ºC. The value of D calculated using equation (2) at temperature T = 273 K was 0.087 cm2 s-1 which is in good agreement with the values given in the literature (Brus et al. (2016), Eisele and Hanson (2000)".*

**Comment 3**

Page 9, condensation sink (CS) calculations: Why did the authors (1) choose to use 303 K in their diffusion coefficient (D) calculation (line 204) and (2) only use the SMPS PNC for the CS (line 203)? In regards to (1): temperature data wasn't reported in this paper but was taken as part of the meteorological data. The historical data reports Beijing's average temperature in January as being around ~270 K. This difference in temperature doesn't lead to a particularly large change in D but certainly is worth addressing.

**Response 3**

(1) We accept the point about the temperature. The average temperature in Beijing during the observations was close to 273K. We have re-calculated our parameters using this value for T. The revised values are given below, and these have replaced the values in the paper.

(2) Regarding the point about calculating the CS, our response to this comment is included in Response 4 below.

**Comment 4**

In regards to (2): I would like to know why the authors chose to neglect the PNC data obtained from the NAIS for <14 nm size bins. A few calculations with "toy" size distributions show that, depending on the number concentration at these <14 nm bins, the CS can be non-trivially changed with the inclusion of these smaller bins. If the size distributions during NPF events in this paper such that the CS hardly changes with the inclusion of the smaller size bins, this should be stated.

**Response 4**

We have re-calculated the parameters, first by holding the temperature at 303K, in order to check if including the particles smaller than 14 nm will make a significant difference to the CS. We found that the CS value increased by about 8% (from 4.8 to 5.2 s$^{-1}$).

Therefore, we have re-calculated all the parameters using the value of CS obtained across the entire size range (< 14 nm from the NAIS plus >14 nm from the SMPS) and with the temperature changed from 303K to 273K.

The original values in the paper are as follows:

$D = 0.092$ cm$^2$ s$^{-1}$
$CS = 4.8 \times 10^{-3}$ s$^{-1}$
$Coag = 8.3 \times 10^{-4}$ s$^{-1}$
$FR = 23$ cm$^{-3}$ s$^{-1}$

The new values obtained are as follows:

$D = 0.087$ cm$^2$ s$^{-1}$
$CS = 4.2 \times 10^{-3}$ s$^{-1}$
$Coag = 7.2 \times 10^{-4}$ s$^{-1}$
$FR = 26$ cm$^{-3}$ s$^{-1}$

We have replaced all the values in the paper accordingly (Page 10, lines 237-240, Section 3.6, Section 3.7 and Table 3).

**Comment 5**

Page 13, lines 294-296: It is also worth mentioning that the pre-existing particles coming into the region from the winds from the south are also increasing the condensation sink, further reducing the likelihood of NPF.

**Response 5**

We have included the following text on page 13, lines 314-315:

*"Pre-existing particles entering the region with the winds from the south will also increase the condensation sink, further reducing the likelihood of NPF."*

**Figures/Tables:**

**Comment 6**

Each figure (excepting Fig 4) could benefit from being more professionally presented. I'm not sure what programming language was used to create these figures but if it is e.g. python, using savefig('name.png',dpi=300) and savefig('name.pdf') would create much nicer looking figures. The text is somewhat blurry and could benefit from being saved at a higher dpi (for png) or as a pdf without the grey backgrounds.

**Response 6**

Since first submission to ACP required embedding the figures within the body of the manuscript, the resolution of the figures has suffered. In the final submission, we shall present each figure as a separate file, so that they will be of much higher resolution.

For now, we have removed the grey background and the frames from all the figures.

**Comment 7**

Figures 2 and 7: The colorbars need labels of units. The numbers on the colorbars are quite blurry and need to be sharpened.

**Response 7**

The numbers that appear on the color bars are produced by the software. As we have provided two labels with the end point values, we feel that these intermediate numbers on the color bars are not essential. To compensate for this, we have further sharpened the text labels at the two ends of the color bars and have added the units ($cm^{-3}$).

**Technical comments:**

**Comment 8**

Abstract, lines 29-31: The sentence would read better is if said 'Estimated characteristics... are very different than to when the measurements. .

**Response 8**

The text has been changed as follows (Abstract, lines 30-32):

*"Estimated characteristics of NPF events, such as their starting times and formation and growth rates of particles, are more accurate when the detection range of particles extends to smaller sizes".*

**Comment 9**

Page 3, line 57: environments (needs an 's')

**Response 9**

The "s" has been added.

**Comment 10**

Page 11, line 247: no comma after 'that'.

**Response 10**

The comma has been deleted.

**Comment 11**

Page 16, line 385: This sentence might read better if it said '...in the smallest particle size bin 2-3 nm for the times at which the rate of increase. . .'

**Response 11**

We agree that the wording is unclear. The text has been changed on page 17, lines 416-418 as follows:

*"...we calculated the formation rate of particles in the smallest particle size bin 2-3 nm. At these times, the rate of increase of particles in this size bin ranged from about $5.0x10^3$ to $1.5x10^4$ $cm^{-3}$ $h^{-1}$."*

**Major Comments 3**

Figure 2, top panel, and Figure 7:
As a consequence, I would argue that particle size distribution data below 2 nm shouldn't even be shown. The concentrations at the size bins <2 nm are subject to instrumental factors, not necessarily resulting from actual variations in the concentrations of sub-2 nm neutral clusters (particles). Hence, their display here could prompt an unaware reader to draw wrong conclusions about the actual population of sub-2 nm neutral clusters.

**Response to Major Comments 3**

In Figure 7, we have excluded the three points below 2.0 nm and inserted a note in the caption cautioning against using the data below 2.0 nm.

Similarly, in the caption to Figure 2(a), we have inserted a note stating that the data below 2.0 nm should be treated with caution.

**Other comments:**

**Comment 1**

Line 68: Kulmama should probably be Kulmala – also in later instances for this reference.

**Response 1**

In the journal paper the name has been mis-spelt as 'Kulmama". Correcting this is bound to affect the citation count and we will seek the advice of the Editor on this matter and make the change, if required.

**Comment 2**

Speaking of which, the recent paper by Kulmala et al. (2017) is relevant to this study and should be brought to attention in the introduction. As condensation sinks were calculated for this study, it might be useful even to shortly discuss the authors' findings in light of the conclusions of that paper (see e.g. lines 237-239).

**Response 2**

We agree. We have inserted the following text into the Introduction (Page 3, lines 69-77):

*"Kulmala et al (2017) proposed that the survival efficiency of clusters to form particles is determined by the two key parameters – condensation sink (CS) and cluster growth rate (GR). They defined a dimensionless survival parameter, P, equal to the ratio $(CS/10^{-4} s^{-1})/(GR/nm h^{-1})$ and showed that P needs to be smaller than about 50 for a notable NPF to take place. However, it was noted that NPF occurred frequently in megacities in China where the calculated P values were much higher. They hypothesized that this discrepancy may be explained if the molecular clusters were being scavenged less effectively than expected based on their collision rates with pre-existing particles or if they grew much faster in size than our current understanding allows".*

In Section 3.8, we estimate the value of P from our results and compare it with the value predicted by Kulmala et al. (2017) for NPF. We have inserted the following text on page 18, lines 442-444:

*"Our values of CS and GR give a cluster survival parameter P = 12 (Kulmala et al, 2017). This value is significantly lower than the maximum value of 50 that was specified as a condition for NPF."*

**Comment 3**

Also, it could be interesting to compare the results here with those in Yu et al. (2016). Therein, they report in particle formation and growth rates during NPF events in Nanjing, also down to sub-3nm sizes.

**Response 3**

We agree. We have included the formation rate and growth rate values found by Yu et al. (2016) in Nanjing by inserting the following text at the end of section 3.7(Page 18, lines 425-429):

*"These values may be compared with that found by Yu et al. (2016) in the urban atmosphere of Nanjing, China. They studied eight NPF events using a nano-condensation nucleus counter system capable of measuring particle size distributions down to 1.4 nm and estimated initial and peak particle formation rates of 210 and 2500 $cm^{-3}\ s^{-1}$, respectively. The formation rates showed good linear correlation with a sulfuric acid proxy".*

And at the end of Section 3.8 (Page 18, lines 441-442):

*"Yu et al. (2016) reported an exceptionally high local maximum growth rate of 25 nm $h^{-1}$ in Nanjing, China.*

**Comment 4**

Lines 277 & Fig. 3, line 287:
"Haze days" seems to be used interchangeably with "no-NPF days". Are they? If so, that point should be made clearer. If not, it may be feasible to mark them in Fig. 3. The various types of day are actually defined later on (lines 325-329). I suggest moving this definition to an earlier place, and then shortly mention it again later.

**Response 4**

In the original Figure 3, the NPF days were shown as red full markers. The points shown in white hollow markers were all other days, including normal (no-NPF) and haze days. We have changed the figure caption to read "other days" instead of "No-NPF Days".

Also, as suggested, we have moved the definition to the methods section and shortly mention it again at this point. The added text in Section 2.3.1 now reads as follows (Page 8, lines 184-188):

*"A day on which there was at least one NPF event as defined above was termed an "NPF day". A day where the above criteria were not fulfilled were classified as a "non-event" day. A "haze day" was defined as a day when the 24-hour average $PM_{2.5}$ concentration exceeded 75 µg $m^{-3}$ - the national air quality standard in China. A day on which there was neither NPF or haze was defined as a "normal day".*

**Comment 5**

Line 318: "attachment to existing particles"

**Response 5**
This sentence was removed when the discussion on cluster concentrations was excluded.

**Comment 6**

I would have expected this process be more pronounced on the *no*-NPF days, when condensation sinks were higher.

**Response 6**

This statement was also removed when the discussion on cluster concentrations was excluded. However, we calculated the condensation sinks for no-NPF days and found that it was 0.006 s$^{-1}$, which is not significantly higher than the corresponding value on NPF days (0.005 s$^{-1}$). However, the condensation sink on haze days was 0.060 s$^{-1}$, which is significantly higher than both normal days and NPF days. We have inserted the following text into the end of section 3.6 on condensation sinks (Page 17, lines 404-408):

*"The value of the condensation sink during NPF events (0.004 s$^{-1}$) was not significantly different to the corresponding average values during other times on NPF days and on normal days with no NPF (0.006 s$^{-1}$). However, the mean condensation sink on haze days (0.060 s$^{-1}$) was significantly higher than both these values."*

**Comment 7**

Line 378: "previous have not been able"
I assume the authors refer to their novel measurement of particles in the 2-3 nm allowing them to more accurately calculating the coagulation sink (CoagS) for particles down to 2 nm. That's technically OK, but one would expect those small particles (i.e. in the 2-3 nm range for instance) to play a minor (negligible?) role in determining CoagS. How much is the value obtained here improved (increased) by the possibility to take the 2-3 nm range into account?

**Response 7**

Equation (5) for the formation rate considers the particles in the size range 2-3 nm. The rate of change of the number of particles in this size range was not available to previous workers. We use this, together with the coagulation sink of the particles in the size range 2-3 nm (CoagSd$_p$) to calculate the formation rate. The coagulation rate CoagS refers to the entire particle size range and this value is, as the reviewer points out, much larger than CoagSd$_p$ in the size range 2-3 nm. However, we thank the reviewer for this comment as it shows that the text was not very clear. We have modified the text as follows to make this as clear as possible:

In Section 2.3.3 (Page 11, lines 254-255) as follows:

*"CoagS$_{dp}$ represents the loss of the particles due to coagulation in the size range 2-3 nm, calculated from equation (4) with d$_p$ = 2 nm, and GR is the growth rate of particles".*

And, in Section 3.7 (Page 17, lines 412-417):

*"Using our value of the CS, we calculated the mean value of the coagulation sink using equation (4) for 2 nm particles during an NPF event to be 7.2x10$^{-4}$ s$^{-1}$. Previous studies in Beijing have not been able to determine this value at 2 nm. The value reported for 3 nm particles for NPF events in Beijing by Wu et al. (2011) was 9.9x10$^{-4}$ s$^{-1}$, which is close to our value at 2 nm. Using our value of the coagulation sink in equation (5), we calculated the formation rate of particles in the smallest particle size bin 2-3 nm".*

**Minor comments:**

**Comment 8**

Abstract, 2nd sentence: The statement should be clarified. From what are the estimated characteristics different in the case of restricted measurements?

**Response 8**

We have modified this sentence as follows (Abstract, lines 30-32):

*"Estimated characteristics of NPF events, such as their starting times and formation and growth rates of particles, are more accurate when the detection range of particles extends to smaller sizes."*

**Comment 9**

Lines 152-153: It may be interesting and instructive for the reader to hear, in short, about the nature of the problems encountered.

**Response 9**

We have replaced this sentence with the following (Page 7, lines 169-170):

*"Data was lost on nine days owing to various problems such as the loss of power, software malfunction and a blocked filter during a haze event."*

**Comment 10**

Line 263: Does this t-test result apply to the whole measurement campaign, or just the subset shown in Fig. 2? In the latter case, would it change when applied to the whole period?

**Response 10**

This was for the subset shown in Fig 2. However, when we consider the entire monitoring period, the corresponding difference was even more significant. We have added the following sentence: (Page 13, lines 303-305)

*"The corresponding difference was even more significant when considering the entire monitoring period where the mean daily values of $PM_{2.5}$ on NPF days and the other days were 21 $\mu g\ m^{-3}$ and 143 $\mu g\ m^{-3}$, respectively."*

**Comment 11**

Line 297: "are more likely …" than what else?

**Response 11**

We have amended the text as follows on page 14, lines 338-431:
*"Thus, the observed haze events are unlikely to be caused by in-situ new particle formation and more likely to be due to particles carried by the wind into the city or being prevented from escaping due to temperature inversions in the atmosphere".*

**Comment 12**

Lines 329-332, Table 2: The source of the uncertainty of 20% has remained unclear to me. Maybe the authors can rephrase.

**Response 12**

We agree that the statement is unclear. We have replaced it with the following text on page 15, lines 361-364:

*"The values shown are the means of the average $PM_{2.5}$ concentrations over all the 24-hour days. The daily mean values varied from day to day, especially on days with NPF events or haze events mainly due to the different durations of these events. We estimated the standard deviation about these mean values to be 20%".*

**Comment 13**

Most figures have a gray background and odd dark-gray or blank frames. They would look better without any that.

**Response 13**

We have removed the grey background and frames around all figures.

**Comment 14**

The text/numbers in the color bar in Figures 2 and 7 are difficult to read and lack units.

**Response 14**

We have improved the quality of the numbers on the color bars and included units ($cm^{-3}$).

[revised manuscript text omitted]